# Effect of Low vs. High Carbohydrate Intake after Glycogen-Depleting Workout on Subsequent 1500 m Run Performance in High-Level Runners

**DOI:** 10.3390/nu16162763

**Published:** 2024-08-19

**Authors:** Tomas Venckunas, Petras Minderis, Viktoras Silinskas, Alfonsas Buliuolis, Ronald J. Maughan, Sigitas Kamandulis

**Affiliations:** 1Institute of Sport Science and Innovations, Lithuanian Sports University, 44221 Kaunas, Lithuania; petras.minderis@lsu.lt (P.M.); viktoras.silinskas@lsu.lt (V.S.); alfonsas.buliuolis@lsu.lt (A.B.); sigitas.kamandulis@lsu.lt (S.K.); 2School of Medicine, University of St Andrews, St Andrews KY16 9TF, UK; ronmaughan@st-andrews.ac.uk

**Keywords:** glycogen depletion, middle-distance race, sports performance, oxidative stress

## Abstract

Carbohydrate (CHO) metabolism is crucial for short-duration, high-intensity exercise performance, but the effects of variations in glycogen availability have not been investigated in field trials of trained athletes. This study was designed to test how 1500 m time trial (TT) performance is affected by the manipulation of pre-race glycogen reserves. Competitive middle-distance runners (*n* = 11 (4 females)) completed a 1500 m individually paced indoor TT after abundant (high, >5 g/kg/d) or restricted (low, <1.5 g/kg/d) dietary CHO intake for 2 days after a glycogen-depleting session. Stride pattern, heart rate (HR), capillary blood lactate, and glucose and plasma malondialdehyde (MDA) response were determined. The TT was slower in low vs. high condition by 4.5 (4.5) s (~2%; *p* < 0.01), with a tendency toward shorter stride length. Blood lactate and glucose were lower before the TT in low vs. high condition (1.8 (0.5) vs. 2.2 (0.7) mmol/L and 5.4 (0.7) vs. 5.9 (0.8) mmol/L, *p* = 0.022 and 0.007, respectively), and peak lactate was higher in high vs. low condition (16.8 (3.1) vs. 14.5 (4.2) mmol/L, *p* = 0.039). Plasma MDA was the same before the TT, and 15 min after the TT, it increased similarly by 15% in low (*p* = 0.032) and high (*p* = 0.005) conditions. The restriction of pre-test CHO intake impaired 1500 m TT performance and reduced baseline and peak blood lactate concentrations but not blood glucose or MDA response.

## 1. Introduction

The ability to perform muscular exercise is affected by the preceding diet. Christensen and Hansen (1939) [1] showed that endurance capacity in prolonged work was enhanced if a diet high in carbohydrate was consumed in the days prior to exercise and was reduced by consumption of a low-carbohydrate diet. Krogh and Lindhard (1920) [2] had previously shown that subjects exercising after a high-carbohydrate diet were less fatigued, and when exercise was preceded by a low-carbohydrate diet, they were more fatigued than after their habitual diet. Soon after this, the benefits of ingesting carbohydrate during exercise were shown in studies of competitors in the Boston marathon race [3]. These studies provided experimental support for the earlier observations of Zuntz (1901) [4] that carbohydrate is a more efficient fuel than fat in terms of oxygen cost: this is important when oxygen availability is a limiting factor in exercise.

The crucial role of carbohydrate availability in exercise performance was confirmed when the needle biopsy technique was applied to investigations of muscle metabolism in the 1960s. Bergström and Hultman (1966) [5] showed that the glycogen content of the exercising muscles is dramatically reduced during prolonged exercise. Hermansen et al. (1967) [6] reported that during cycling exercise at a work rate corresponding to approximately 75% of maximum oxygen uptake (*V*O_2_ max), marked depletion of the glycogen stores of the quadriceps muscles occurred at exhaustion. Bergström and Hultman (1966) [5] had already shown that consumption of a diet rich in carbohydrate for a few days after exercise-induced glycogen depletion resulted in a rapid resynthesis of the glycogen stores; after 2–3 days on a high-carbohydrate diet, muscle glycogen content, in those muscles which had been exercised, was about 2–3 times greater than the resting value. If a low-carbohydrate diet is consumed in the days after exercise-induced glycogen depletion, the muscle glycogen content remains low for several days [7]. Thus, by using a combination of exercise and dietary modification to manipulate the glycogen content of the muscles, Bergström et al. (1967) [8] were able to show a close relationship between the pre-exercise muscle glycogen content and the endurance time during cycling exercise at about 70–75% *V*O_2_ max. The obvious conclusion from these studies was that the availability of carbohydrate in the form of liver and muscle glycogen stores represents a limiting factor in this type of exercise, and although there might be some qualifications, this remains generally true. A second conclusion is that performance is strongly influenced by the pre-exercise diet and exercise regimen.

By contrast, the cause of fatigue in high-intensity exercise of a short duration (a few minutes) has been less studied and is not clearly understood, though many different mechanisms have been proposed. Here, fatigue can be defined as the inability to maintain the expected or required power output. The rate of muscle glycogen degradation during exercise increases exponentially with respect to exercise intensity [9], but the duration is necessarily short when the intensity is high. Saltin and Karlsson (1971) [10] and Hermansen (1981) [11] concluded that the availability of muscle glycogen does not normally limit endurance capacity at work rates in excess of about 90% *V*O_2_ max. The normal muscle glycogen content is about 100 mmol glucosyl units kg^−1^ wet weight (w.w.) [12]. During exercise at 100% *V*O_2_ max, the glycogen depletion rate is about 11 mmol glucosyl units kg^−1^ w.w·min^−1^ [13]. Endurance time at this work rate is generally in the order of 3–6 min, so there should be adequate glycogen available.

Some researchers (e.g., [14]) have measured muscle glycogen concentration at the point of fatigue during intense cycling exercise of a short duration and have reported values in excess of 50 mmol glucosyl units kg^−1^ w.w. This value is well above the *K*_m_ for phosphorylase and suggests that substrate availability should not be limiting. These values relate to the whole muscle, however, and it is of course possible that with low pre-exercise muscle glycogen stores, high-intensity exercise performance could be limited by glycogen depletion in a few large motor units and most likely in some fast-twitch (Type II) fibers. With normal pre-exercise glycogen levels, the glycogen content of Type II fibers during repeated sprints decreases sooner and to a greater extent than that of Type I fibers [15], but over 60% of the initial pre-exercise muscle glycogen content was still left in the Type II fibers at exhaustion.

More recently, it has been shown that the total muscle glycogen content might be less relevant than the specific pools of glycogen located at different sites within the muscle cell [16]. Key cellular functions, including those of the sodium–potassium and calcium pumps, the calcium-release channels, and the proteins that interact to generate force in the muscle fibers, occur at specific locations within the muscle cell. There are also specific pools of glycogen molecules, and studies from Ørtenblad et al. (2011) [17] and Gejl et al. (2014) [18] have shown that the intramyofibrillar pool may be particularly important during high-intensity exercise. Preferential utilization of this storage pool occurs and seems to be correlated with an impaired release of calcium ions from the sarcoplasmic reticulum. This in turn leads to impaired excitation–contraction coupling within affected muscle fibers that may contribute to, or even be the cause of, fatigue during high-intensity exercise.

In spite of the evidence supporting a potential role for glycogen availability as a determining factor in the performance of high-intensity exercise, relatively few studies are available to provide experimental support. Maughan and Poole (1981) [19] showed that exercise time to exhaustion on a cycle ergometer at a work rate equivalent to 105% *V*O_2_ max was affected by a pattern of diet and exercise intended to manipulate pre-exercise glycogen availability: exercise time after normal diet was 4.87 ± 1.07 min; after a low-carbohydrate diet, it was reduced to 3.32 ± 0.93 min, and after a high-carbohydrate diet, endurance time was longer than on the normal diet (6.65 ± 1.39 min). Pizza et al. (1995) [20] also found that glycogen loading above normal levels with abundant CHO intake can augment laboratory-based exercise cycling performance in tests lasting a few minutes. Notwithstanding, most laboratory studies have found no effect of this glycogen supercompensation on exercise of a short duration [21,22,23,24,25,26]. In addition, the laboratory evidence of possible improved physiological function or ergometer performance does not necessarily translate to real-world scenarios [27].

The main aim of the present study, therefore, was to test whether dietary carbohydrate restriction after glycogen-depleting exercise affects subsequent middle-distance time trial performance in well-trained runners. It was hypothesized that perturbations in glucose metabolism could have negative consequences on racing ability. In addition to racing performance, gross running biomechanics, and physiological responses, an oxidative stress biomarker, plasma malondialdehyde (MDA), an end product of lipid peroxidation, was also measured. The latter addressed the hypothesis that relative glucose deprivation within working muscles under conditions of low CHO availability induced higher oxidative stress.

## 2. Methods

### 2.1. Participants

Eleven (4 females) well-trained competitive middle-distance runners (mean (SD) age 21 (4) y, height 180 (11) cm, BMI 21 (2) kg/m^2^, and training experience 8 (range 3 to 13) y) volunteered to participate in this study. This study was performed during the athletes’ cross-country racing season (spring), and during the last 4 months, participants were competing indoors until about 2 months before the start of the study, when they switched to largely outdoor training and preparation for cross-country races. Their mean weekly training volume (calculated as the average over the previous month) was 12.3 (1.8) h and included 69 (25) km of running. The personal best IAAF score of the whole group was 841 (103) on average [men 779 (24), range 750–827, women 950 (98), range 812–1040], which would correspond to tier 3, as suggested to classify the level of athletes recently [28]. Exclusion criteria were illness or injury that precluded normal regular training for longer than a week during the previous 6 months; dietary manipulations such as energy restriction or avoidance of carbohydrates; 1500 m personal best time slower than 4:20 for male and 5:00 for female participants.

All the participants read a description of the study before providing their written informed consent for voluntary participation. The study was conducted in alignment with the recent update of the Declaration of Helsinki and was approved by the Lithuanian Sports University Biomedical Research Ethics Committee (No. TRS(M)-29912, 1 February 2024).

### 2.2. Organization of the Study

This study was a repeated-measures randomized crossover design study, with participants completing a 1500 m time trial (TT) on an indoor 200 m running track after abundant (high) or restricted (low) dietary CHO consumption for 2 days after a glycogen-depleting training session (GDS). Time trials were performed 4 weeks apart, with training and diet during the study period, and especially during the last week preceding the TT, kept as similar as possible. Both the GDS and TT were performed on the same indoor 200 m track in the afternoon at always about the same individual time for the participant.

### 2.3. Diet Analyses and Manipulations

Habitual diet before each period of diet manipulation was recorded daily for the three consecutive days before the GDS by weighing foods, supplements, and beverages and recording intake in a 24 h food diary; a photographic record was made (for verification) of everything consumed. Participants were instructed to email their food tracking data either at the end of the same day or the next morning. This timely submission enabled the researchers to monitor food intake in real time, to clarify any ambiguous entries, and to provide feedback to improve the accuracy of the collected data. Food data were collected and analyzed by one of the researchers (PM), who calculated energy and macronutrient intake using the Cronometer program (https://cronometer.com/, accessed on 1 March 2024). Entries for the foods reported by the participants were selected from institutional databases, including the NCCDB (Nutrition Coordinating Center Food & Nutrient Database) and USDA SR28 (United States Department of Agriculture National Nutrient Database for Standard Reference). For the two days between the GDS and the TT, participants either ate >5 g/kg (aiming for at least ~10 g/kg to meet the recommended amount for CHO loading (which is currently agreed to be 10–12 g/kg/d for 36–48 h [29]; high CHO intake)) or <1.5 g/kg (aiming for ~1 g/kg) of CHO per day (low CHO intake).

Since the study employed a crossover design, participants underwent both dietary manipulations with a 4-week washout period in between; the order of treatment allocation was randomized. Before starting the dietary CHO manipulation, participants were thoroughly instructed for practical recommendations on how to modify their diet to achieve the desired dietary intake until the TT. To keep the diets after the GDS until the TT either low or high in CHO, participants on the low-CHO intake trial were required to substitute most of their habitual CHO intake by fat, while the high-CHO intervention required consuming foods high in CHO and low in fats. The amount of protein intake was aimed to be unchanged. Participants were contacted at least once daily by the same researcher to make sure that the recommendations were clear, and the dietary log and food snapshots were checked at the same time to make sure that the instructions were followed. If needed, additional counseling and corrections were made. The diet analysis for major macronutrients (carbohydrates, fats, and proteins) and water content (from beverages and food) was conducted in the same manner as the habitual food intake assessment. Dietary parameters were expressed as absolute values and normalized to participants’ body mass.

### 2.4. Glycogen-Depleting Exercise Session (GDS)

A glycogen-depleting exercise session (GDS) was conducted on an indoor 200 m running track 48 h before each TT and comprised 60 min of running at 66% of the intended 1500 m TT speed followed a few minutes after by 10 sets of 200 m (with 200 m jog for recovery) at 1500 m TT pace. A similar training session using stationary cycling was shown in our previous study of recreationally active participants to reduce the *m. vastus lateralis* glycogen content to very low levels [30]. To reduce gluconeogenesis from lactate produced by the 10 × 200 m sprints and to minimize glycogen replenishment at least in Type I muscle fibers [31], 15 min of jogging at 50% of 1500 m TT speed was completed immediately after the last 200 m bout. Participants were supervised during the GDS and were wearing either their own HR meter or one provided by the researchers. Blood lactate was measured after the 60 min of the steady-state run, after the intervals, and then after 15 min of jogging.

During the next day after the GDS, a 30 min running session at 66% of the planned 1500 m TT speed was completed to help keep glycogen levels low in the low-CHO trial and at the same time to allow for the nearly maximal accumulation of muscle glycogen until the TT for the high-CHO trial. No additional training and no caffeine, alcohol, or supplements were allowed within 48 h of the TT.

### 2.5. 1500 m Time Trial (TT)

Participants arrived around midday at the indoor track and field arena after having their last meal 2 to 5 h before (following their habitual pre-competition eating pattern). After visiting the toilet, their nude body mass was measured and body composition was estimated by bioelectric impedance analysis (Tanita TBF-300, Tokyo, Japan). The athletes then warmed-up by jogging for 10 min at their usual comfortable easy pace and performing 10–15 min of dynamic stretching, skipping, short strides, etc., as part of their standard routine. Before starting, an HR sensor (Polar H10, Kempele, Finland) was securely strapped on the chest and an HR monitor was attached on the wrist for continuous measurements of HR during the TT. Capillary blood lactate and glucose concentrations were measured ~1 min before the TT, and the instructions were repeated.

Runners started the 1500 m individual TT at the half-way point of the arena after being instructed to complete the TT in the fastest possible time. A custom-made red lamp leader installed on the balcony of the arena was pre-set at the intended (pre-arranged) pace of the participant, which was agreed between the coach and the runner based on the current capacity. It was not obligatory to follow the pacing lamp, but runners were recommended to keep with the lamp for at least the initial 400 m and then decide on their further pacing strategy depending on how they were feeling. Timing gates (Witty, Microgate, Bolzano, Italy) were placed at the start and finish lines, and thus interim times were automatically recorded every 100 m. The researchers, who were blinded to the dietary intake data, cheered and encouraged the runners with every lap completed. The time trial was run individually with spikes and competitive attire. The ambient temperature in the arena was 19.5 to 20.5 °C, and the relative humidity was 45 to 55%.

Gross running mechanics (running spatiotemporal parameters) were measured by placing 10 m light beam gates (the Optojump system, Microgate, Bolzano, Italy) on the first lane 50 m from the finish line to record the step rate and length and contact and flight phase durations during each 200 m lap. Analysis of the data showed no consistent pattern of change over time, so the values of each parameter were averaged across all eight laps.

After crossing the finish line, participants lay down on a soft mat for 15 min of passive recovery with continued HR recording, and fingertip blood lactate (Lactate Pro2, Arkray, Kyoto, Japan) and glucose (Ascensia Contour Plus, Bayer Healthcare AG, Leverkusen, Germany) levels were measured at 1, 3, 5, and 15 min post-exercise. Recovery jogging was then allowed as desired. Within 1 min after the completion of the TT, participants were asked to rate their average (not peak) efforts during the TT. The 6–20-point Borg scale of rating of perceived exertion (RPE) was used for this purpose [32].

### 2.6. Malondialdehyde (MDA) Measurement

Blood samples were taken by venous puncture from an ante-cubital vein upon arrival (before the warm-up) and then 15 min after the TT, dispensed into EDTA-containing tubes and kept up to 2 h on ice until harvesting plasma by 1500× *g* centrifugation for 15 min at 4 °C. Then, plasma aliquots were stored at −80 °C until the analysis of MDA. Spectrophotometric analysis (Spark 10M Tecan, Männedorf, Switzerland) of the samples was carried out at 450 nm wavelength to measure MDA concentration using commercially available kits (Ref #E1371Hu, BT Lab, Zhejiang, China) by following the provided instructions. To ensure accuracy, samples were analyzed in duplicate, and average values were reported. The coefficient of variation of the assay was <10%.

### 2.7. Statistics

The data are presented as mean and standard deviation (SD). The Shapiro–Wilk test was used to confirm that the data were normally distributed. MDA values were not distributed normally; therefore, the non-parametric Kruskal–Wallis test was used to test for MDA level changes and differences between the conditions. The differences in body mass and dietary intake before the GDS and during the 2-day period after the GDS preceding the 1500 m TT, as well as the differences in 1500 m and interim times, rating of perceived exertion, gross stride pattern parameters, and blood lactate and glucose responses to the 1500 m TT run, were assessed using a two-way repeated-measures ANOVA, one-way repeated-measures ANOVA, or paired *t*-tests as appropriate. When the ANOVA showed a significant effect, post hoc tests with Sidak correction for multiple comparisons were performed. Effect sizes (ESs) for pairwise comparisons were assessed using Cohen’s d for parametric analyses and interpreted as trivial (d < 0.20), small (0.20–0.59), moderate (0.60–1.19), large (1.20–1.99), or very large (≥2.0) [33]. For non-parametric analyses, ES was calculated using the partial eta squared (ηp^2^) coefficient and interpreted as follows: small, 0.01–0.05; moderate, 0.06–0.13; and large, ≥0.14. Pearson correlation was used to test for a relationship between CHO intake in the high-CHO condition and overall TT time change over that after the low-CHO condition. Statistical significance was set at *p* < 0.05. Statistical analyses were performed using GraphPad Prism version 10.2.3 (403) for Windows (GraphPad Software, Boston, MA, USA).

## 3. Results

### 3.1. Diet and Body Mass

None of the athletes appeared to be vegetarians, and all athletes succeeded with the manipulation of the dietary CHO intake (Table 1). Dietary CHO intake measured during the 3 days before each glycogen-depleting session was moderate and averaged 4.8 (1.4) g/kg/d (net CHO intake, i.e., excluding dietary fiber, was 4.4 (1.5) g/kg/d), which comprised ~48% of the total energy intake (fat contributed ~36% and protein the remaining ~16%).

During the low condition, CHO intake was <1.5 g/kg/d in all runners (average 1.0 (0.4) g/kg), and during the high-CHO condition, it was >5 g/kg/d in all runners (average 9.7 (2.6) g/kg/d) (Table 1). Compared to the low-CHO condition, a high CHO intake resulted in 0.7 kg higher body mass (*p =* 0.048), which was 0.9 kg higher (*p* < 0.001) than at baseline with habitual dietary intake (Table 1). CHO intake and total energy intake were increased on the high-CHO trial compared to both the usual intakes and intake before the TT on the low-CHO diet. In addition, total water intake was higher and protein intake was lower on the high- compared to the low-CHO diet (Table 1).

### 3.2. Glycogen Depletion Session

Heart rate during the 60 min of steady-state running phase during the GDS averaged 161 (8) bpm, and the post-exercise blood lactate concentration was 1.7 (0.8) mmol/L. The average time of the 200 m intervals was 35.1 (2.5) s, peak heart rate was 183 (4.6) bpm, and lactate within 1 min after the last interval was 8.3 (3.2) mmol/L. After 15 min of recovery jogging after the completion of the 200 m intervals phase, lactate was 1.5 (1.4) mmol/L. There were no differences between the two conditions in these and any other parameters measured during the training GDS (*p* > 0.05).

### 3.3. Time Trial

All participants completed two TTs separated by four weeks. The time to complete the first trial—irrespective of diet—was 278.3 (17.8) s, while the time to complete the second TT was 277.1 (18.1) s (*p* = 0.617). There was also no order and diet interaction effect (*p* = 0.811). Time to complete the 1500 m TT was longer in the low- vs. high-CHO condition by 4.5 s (~2%) on average (*p* = 0.009), which was due to lower running velocity, especially toward the end of the run (Table 2).

There was a significant time (lap) effect for three of the four spatiotemporal running parameters measured, with step length (*p* = 0.035) and flight phase time (*p* < 0.001) decreasing, while stance phase time increased (*p* < 0.001) with laps accumulated. However, there was no order (irrespective of the diet) effect or order and diet interaction effect for these parameters on gross biomechanical stride pattern. Despite significantly lower body mass after the low-CHO diet (Table 1), the average stride length (*p* = 0.086), step frequency (*p* = 0.586), flight phase (*p* = 0.186), and stance phase times (*p* = 0.323) were not different between trials (Table 2). There was no correlation between the CHO intake (in g/kg/d) in the high-CHO condition and time improvement over the low-CHO condition. Peak heart rate response to the TT was the same across the conditions (low CHO 188 (9), high CHO 189 (5) bpm), while the rating of the perceived exertion was significantly higher after the low-CHO diet (Table 2).

The time trial elicited a marked increase in blood lactate and glucose concentrations, as well as a 15% increase in plasma MDA concentration (Table 3). After the high-CHO diet, the blood lactate concentration was higher (*p* < 0.05) before and 1 min after the TT (Table 3). The peak blood lactate concentration was higher in the high-CHO condition (*p* = 0.039), but the lactate increase from baseline (delta, or response) and the recovery of lactate during the 15 min of passive recovery were not different between conditions (*p* > 0.05) (Table 3). The blood glucose concentration was higher before the TT in the high-CHO condition (*p* = 0.007), but post-race response (delta) and recovery during the 15 min of recovery were similar in both conditions (*p* > 0.05). The plasma MDA level did not differ between dietary conditions before the TT, but 15 min after the TT, it increased similarly in the low (*p* = 0.032)- and high (*p* = 0.005)-CHO condition (Table 3). Female runners (*n* = 4) had higher (*p* < 0.05) MDA levels, but the response of both sexes was the same.

## 4. Discussion

The main finding of the present study was that two days of restricted carbohydrate intake after exercise designed to reduce muscle glycogen content impaired the subsequent 1500 m time trial performance of well-trained middle-distance runners compared to the high-carbohydrate trial. Pre-exercise blood lactate and glucose concentrations and peak lactate were slightly lower with low CHO intake. There was also a higher rating of perceived exertion with carbohydrate restriction compared with copious amounts of dietary CHO for 2 days preceding the time trial.

The primary aim of most recent laboratory studies [26,30,34] has been to attempt to decipher the mechanisms underlying the effects of the manipulation of glycogen availability rather than measuring effects on performance, and the exercise models used seldom represent a real-life sports situation. In the context of the current study, there are well-designed laboratory studies such as by Maughan and Poole [19] and Schytz et al. [26] that focus on time to exhaustion at a fixed power output or work carried out during a fixed amount of time, respectively, rather than measuring the fastest time to complete the distance as in nearly all competitive sport situations. These studies also normally use recreationally active participants who are not familiar with the exercise task involved rather than well-trained athletes. Those studies have, therefore, limited relevance for practitioners.

The results of our study agree with the findings of early studies that CHO restriction reduces time to exhaustion in cycle ergometer exercise lasting a few minutes in the laboratory setting [19,35]. They are, however, in disagreement with the findings of no performance effects of a more recent study where moderately trained runners performed four time trials over distances of 50, 400, 1500, and 3000 m after acute manipulation of carbohydrate availability [25]. The results of that study, though, are not easily interpreted: the 50 m sprint was performed on an indoor track but the others on a motorized treadmill, with only a short interval (5–10 min) between runs; no familiarization to the high-speed treadmill running tests was allowed. The participants’ CHO intake for the ad libitum diet phase during the 3 days preceding the GDS phase was close to 5 g/kg/d on average; in about half of the participants, intake was below the range of 5–8 g/kg/d recommended for athletes training 5–6 times per week [36]. This could be primarily because our participants were not training by accumulating high training loads during this phase of the study and especially the days preceding the TT. During the high-CHO feeding phase, the CHO intake of all the athletes was above 5 g/kg/d, and all but one had a CHO intake between 6.8 and 13.4 g/kg/d, which is within the recommended range for CHO loading [37]. Muscle glycogen content was not measured in this study, but the reduction in body mass by an average of 0.9 kg between the pre-GDS and pre-TT after the CHO restriction with the low-CHO diet is consistent with a reduction in muscle glycogen stored and the associated muscle water content [38]. A very similar body mass change was reported in another well-controlled study, where high CHO intake was compared with moderate CHO intake [39].

Protein intake for the 2 days preceding the TT was slightly lower on the high-CHO trial, and water consumption was slightly lower on the low-CHO trial, but these differences were generally small and unlikely to affect performance. Even if some participants fell slightly short (both CHO conditions) of the recommended protein [40] and fiber [41] intake, it is unlikely that this short-term relative deprivation had any effect on their TT performance, especially since they had two days of recovery after the GDS. Large variations in dietary protein intake will affect acid–base status [42] and may be important in 1500 m running, where a profound metabolic acidosis develops. Experimental evidence, however, does not support this hypothesis [43]. The design of our study does not allow for a clear delineation of whether the low CHO intake after the GDS compromised the performance or if it was rather the high-CHO intake which allowed for improved running capacity. Previous studies, however, suggest that the former is more likely [21,22,23,24,35,44]. In any case, our study clearly suggests that the availability of CHO before middle-distance racing seems of importance for overall performance. The restriction of CHO intake might reduce blood glucose levels, which may hinder the central drive and diminish performance ability [45,46], but the present study showed no evidence of the development of hypoglycemia in the low-CHO condition. The pre-TT blood glucose level was slightly lower in the low-CHO condition, but elevated baseline glucose and lactate on the high-CHO trial might simply reflect the last meal effect; the post-exercise blood glucose concentration was the same on the two treatments.

The tendency for a decrease in step length rather than step frequency despite significantly reduced body mass in the low-CHO condition indicates a somewhat more pronounced decline in propulsive power with smaller muscle glycogen reserves. This was most likely due to the earlier fatigue of initially recruited Type II muscle fibers and then gradual switching to the recruitment of more Type I fibers (or the other muscles, slightly changing the running technique and lowering movement economy), which precluded the maintenance of the pace. Even if initially recruited Type II muscle fibers do not become completely depleted of glycogen, their input to the muscle contraction and propulsion during running might decrease because of the critical reduction in glycogen in specific locations (distinct compartments) of the fiber, the intramyofibrillar glycogen pool in particular [17,47,48,49,50,51,52]. In support, reduced muscle glycogen content with dietary CHO restriction during an intense training period was shown to be associated with muscular fatigue and reduced running efficiency [53]. Lowered glycogen reserves have been shown to impair sarcoplasmic reticulum calcium cycling during exercise after diets with CHO contents similar to those in the present study [54]. In well-trained skiers, recovering glycogen reserves after a 1 h race was associated with the recovery of sarcoplasmic reticulum calcium turnover [17].

It has been shown recently that regular high-intensity interval training performed with maximal efforts increases baseline plasma MDA levels [55]. This is consistent with the observation that plasma MDA concentration increases in response to all-out exercise in the normal (but not over-trained) state [56]. The female subjects in the present study had higher baseline plasma MDA levels than the male subjects, and this has previously been shown by others [57], but the underlying mechanisms are not clear. It should be noted, though, that a decrease in plasma volume by about 15% has been reported 15 min after exhaustive exercise (even of a longer duration) [58], which is about the same as the 15% increase in the plasma MDA concentration observed 15 min after exercise in our study. In addition, the same study reported that the decrease in plasma volume was less pronounced in a glycogen-depleted state. In the current study, there was no effect of the preceding diet on the baseline plasma MDA concentration or in the post-exercise value. This agrees with the results of Psilander et al. [59], who found no difference between low vs. high initial muscle glycogen conditions for hydrogen peroxide and MDA production within the muscle 3 h after 80 min of cycling at 64% *V*O_2_ max. Finally, this is somewhat in analogy with the absence of an effect of post-exercise carbohydrate restriction on the muscle acute response of metabolic gene expression [60], implying that acute CHO manipulation either before or after exercise associated with the depletion of muscle glycogen does not augment the induction of oxidative stress.

### Limitations

The primary aim of this study was to assess the effects of manipulating carbohydrate status on performance; limited additional measurements were made to limit the burden on the participants and to reduce distractions. Muscle glycogen content was not measured, but a study with a similar protocol using cycling exercise resulted in nearly complete exhaustion of the glycogen content of the active muscles [30]. With the subsequent feeding of only ~140 g CHO per 2-day period until the TT on the low-CHO trial and an additional training session of 30 min between the GDS and TT, where at least half of that content of CHO would have been oxidized, we are sure that the glycogen reserves before the TT were quite different between the feeding conditions.

To avoid participants being influenced by tactical considerations or the performance of others, the exercise test used was not a true competition but rather a simulated race for the fastest time possible that day. However, peak lactate values were similar to those recorded after 1500 m races [61], suggesting that a maximal effort was achieved. While many middle-distance races in championships are run tactically for medals but not necessarily personal best times, there are many other competitions when runners strive not for high places but for fast times and records to be broken. Both the ability to change speed along the distance (including acceleration for a dash to finish) and capacity to maintain high constant speed over the entire distance are important for success in distance running.

The total number of participants recruited for the study was relatively small, but this is common practice in nutritional studies involving high-caliber athletes [18,62,63]. To ensure that the conclusions are applicable to well-trained athletes, for the purpose of the study, we preferred to have a high recruitment standard in terms of the 1500 m run performance and were therefore limited regarding the pool of participants. The small number of female participants precludes definite conclusions on possible sex differences in the response to short-term CHO manipulation. In addition, the menstrual cycle phase was neither assessed nor controlled, but the two conditions were separated by 4 weeks, which is roughly the duration of the normal menstrual cycle. However, despite markedly different plasma MDA levels and TT performance as well as a relatively higher level of female athletes (based on IAAF scores for personal best times), no obvious between-sex differences were seen.

To make the two interventions more contrasting, the CHO intake for the high-CHO condition was increased above that consumed habitually, which resulted in doubling of CHO intake for 2 days before the TT. Therefore, there is no certainty that it was not an increased CHO intake that resulted in superior running ability rather than the restriction of CHO that decreased the performance. However, given the relatively short nature of the TT, the latter seems more likely. Importantly, there was no correlation between the CHO intake in the high-CHO condition (as g/kg/d) and time improvement over the low-CHO condition. In addition, this implication is supported by the majority of similar laboratory-based cycle ergometer studies [21,22,23,24,26,35].

As with all dietary assessments, the results of the baseline diet should be treated with caution: awareness of this assessment and a desire to under-report some (e.g., junk) foods or choose “healthier” products may have distorted the usual dietary intakes. Nonetheless, this initial phase of diet analysis was not essential for the study aims but was rather helpful for a better control of the intakes during the last 2 days before the TT.

Finally, by not blinding participants to the diet with the current design of the study, there is no certainty that there was or was not a placebo/nocebo effect of altered CHO intake, but conversations with the runners gave the impression that there was no disproportional bias toward either of the diets. In addition, awareness of the CHO amount ingested has not been found to be of practical importance by others [20].

## 5. Conclusions

Compared to carbohydrate restriction, a diet with ample carbohydrate provision after a glycogen-depleting session allowed for a faster 1500 m performance in well-trained middle-distance runners, and that was despite the higher body mass compared with the carbohydrate-restriction condition. The restriction of dietary carbohydrate to <2 g/kg/d after the glycogen exhaustion session not only decreased middle-distance racing performance after two days but also increased perceived exertion and slightly reduced baseline and peak blood lactate, while it did not change blood glucose or plasma MDA response to the 1500 m time trial.

## Figures and Tables

**Table 1 nutrients-16-02763-t001:** Body mass and dietary intake before glycogen depletion session (GDS) and during the 2-day period after GDS preceding the 1500 m time trial (TT) on low- and high-CHO feeding regimens.

	Low-CHO Diet	High-CHO Diet	Difference between Diets	
			*p* Value	Effect Size (Cohen’s d)
Body mass before GDS, kg	68.6 (11.3) *[53.1–87.4]*	68.4 (11.4) *[53.4–87.7]*	0.166	0.02
Habitual (Baseline) dietary intake ^a^				
Energy intake, kcal/d	2789 (621) *[2192–3927]*	2724 (671) *[1623–3872]*	0.556	0.10
Energy intake, kcal/kg/d	41.1 (8.5) *[26–54]*	39.7 (6.8) *[25–50]*	0.452	0.18
CHO, g/d	333 (128) *[186–612]*	328 (119) *[188–580]*	0.674	0.04
CHO, g/kg/d	4.9 (1.5) *[2.3–7.3]*	4.8 (1.3) *[3.0–6.9]*	0.578	0.07
Fiber (indigestible CHO), g/d	25.5 (7.6) *[15.3–37.2]*	28.0 (7.3) *[13.8–39.4]*	0.289	0.33
Net CHO, g/d	306 (127) *[147–578]*	299 (120) *[147–548]*	0.503	0.06
Net CHO, g/kg/d	4.4 (1.5) *[2.0–6.8]*	4.3 (1.4) *[2.6–6.6]*	0.408	0.07
Fat, g/d	113.5 (22.3) *[83–151]*	111.2 (25.4) *[61–138]*	0.808	0.10
Fat, g/kg/d	1.8 (0.5) *[1.2–2.7]*	1.6 (0.4) *[0.9–2.3]*	0.692	0.20
Protein, g/d	120.1 (25.6) *[73–153]*	114.2 (28.1) *[81–167]*	0.541	0.22
Protein, g/kg/d	1.8 (0.4) *[1.0–2.6]*	1.7 (0.3) *[1.3–2.3]*	0.434	0.25
Water, L/d	3.2 (1.5) *[1.5–5.9]*	3.3 (1.4) *[1.5–5.9]*	0.701	0.07
Water, mL/kg/d	47.5 (24.0) *[22–89]*	48.6 (21.1) *[23–84]*	0.755	0.05
Dietary intake for 2 days preceding TT				
Body mass before TT, kg	67.7 (11.0) *[52.9–85.4] ****	68.4 (11.4) *[53.9–87.1]*	**0.048**	**0.07**
Energy intake, kcal/d	2595 (680) *[1558–4075]*	4008 (1014) *[2940–5948]* **	**<0.001**	**1.66**
Energy intake, kcal/kg/d	38.5 (9.5) *[23–61]*	59.0 (13.1) *[38–83] ****	**<0.001**	**1.83**
CHO, g/d	69 (25) *[37–115] ****	655 (184) *[358–941] ****	**<0.001**	**5.58**
CHO, g/kg/d	1.0 (0.4) *[0.5–1.5] ****	9.7 (2.6) *[5.4–13.4] ****	**<0.001**	**5.80**
Fiber (indigestible CHO), g/d	19.1 (8.1) *[5.1–30.5] **	33.4 (10.4) *[7.2–45.5] **	**<0.001**	**1.55**
Net CHO, g/d	50 (20) *[27–91]*	621 (177) *[350–895]*	**<0.001**	**5.80**
Net CHO, g/kg/d	0.7 (0.3) *[0.4–1.1]*	9.2 (2.4) *[5.3–12.7]*	**<0.001**	**6.30**
Fat, g/d	202.3 (56.8) *[117–326] ****	114.0 (42.0) *[64–204]*	**0.001**	**1.80**
Fat, g/kg/d	3.0 (0.8) *[2.1–4.9] ****	1.7 (0.6) *[0.7–2.8]*	**0.001**	**1.86**
Protein, g/d	137.4 (45.1) *[47–208]*	108.7 (21.6) *[72–150]*	**0.038**	**0.87**
Protein, g/kg/d	2.1 (0.7) *[0.8–3.1]*	1.6 (0.3) *[1.1–2.2]*	**0.044**	**1.00**
Water, L/d	2.7 (1.4) *[1.0–5.0] **	3.4 (1.5) *[1.8–5.6]*	**<0.001**	**0.48**
Water, mL/kg/d	41.5 (23.7) *[15–90] **	50.7 (23.4) *[27–98]*	**<0.001**	**0.39**

Notes. Data are mean (SD) *[range]*. Significant effects of the diet are highlighted as bold *p* values and effect sizes. CHOs, carbohydrates; TT, time trial. *^a^* estimated during 3 days before GDS. ** p* < 0.05, *** p* < 0.01, **** p* < 0.001 compared to habitual intake.

**Table 2 nutrients-16-02763-t002:** The 1500 m and interim (split) times, rating of perceived exertion, and gross stride pattern (spatiotemporal running parameters).

	Low-CHO Diet	High-CHO Diet	*p* Value	Effect Size (Cohen’s d)
1500 m time, s	279.9 (16.9)	275.4 (18.7)	0.009	0.98
100 m split, s	17.9 (1.2)	17.6 (1.4)	0.173	0.23
200 m split, s	35.9 (2.4)	35.7 (2.7)	0.498	0.08
400 m split, s	72.0 (4.8)	71.7 (5.5)	0.586	0.06
600 m split, s	108.5 (6.9)	107.9 (8.2)	0.462	0.09
800 m split, s	145.8 (8.7)	144.6 (10.7)	0.280	0.12
1000 m split, s	184.0 (10.7)	182.0 (13.0)	0.104	0.17
1200 m split, s	223.1 (13.7)	220.1 (15.4)	0.087	0.20
1400 m split, s	261.3 (16.3)	257.5 (17.8)	0.061	0.22
Rating of perceived exertion	16.9 (2.1)	15.7 (2.0)	**<0.001**	**0.56**
Flight phase time, ms	133.1 (5.0)	135.0 (4.8)	0.186	0.39
Stance phase time, ms	188.9 (8.1)	187.6 (8.0)	0.323	0.12
Step frequency, /min	186.8 (1.9)	186.4 (2.1)	0.586	0.19
Step length, cm	176.6 (5.3)	178.5 (5.1)	0.086	0.36

Notes. Data are mean (SD). Significant effects of the diet are highlighted as bold *p* values and effect sizes.

**Table 3 nutrients-16-02763-t003:** Blood lactate, glucose, and malondialdehyde (MDA) response to the 1500 m TT run.

	**Low-CHO Diet**	**High-CHO Diet**	**Difference between Diets**	
			***p* Value**	**Effect Size (Cohen’s d)**
Lactate, mmol/L				
Baseline (pre-TT)	1.7 (0.5)	2.2 (0.7)	**0.022**	**0.82**
1 min post-TT	14.1 (3.6) ***	16.0 (2.6) ***	**0.047**	**0.60**
3 min post-TT	13.5 (4.5) ***	15.1 (4.1) ***	0.059	0.37
5 min post-TT	12.9 (4.1) ***	14.4 (3.1) ***	0.370	0.41
15 min post-TT	9.5 (3.7) ***	11.4 (3.4) ***	0.116	0.53
Peak value	14.5 (4.2) ***	16.7 (3.1) ***	**0.039**	**0.59**
Delta (response)	12.7 (3.8)	14.5 (2.6)	0.072	0.55
Delta (fold)	8.4 (1.8)	8.0 (1.9)	0.672	0.21
Glucose, mmol/L				
Baseline (pre-TT)	5.4 (0.7)	5.9 (0.8)	**0.007**	**0.67**
1 min post-TT	7.8 (1.4) ***	7.5 (1.3) ***	0.795	0.21
3 min post-TT	8.7 (1.2) ***	8.8 (1.5) ***	0.138	0.07
5 min post-TT	8.5 (1.1) ***	9.0 (1.5) ***	0.182	0.38
15 min post-TT	7.8 (1.3) ***	8.0 (1.2) ***	0.618	0.16
Peak value	8.7 (1.2) ***	9.0 (1.4) ***	0.193	0.23
Delta (response)	3.3 (0.7)	3.1 (1.0)	0.466	0.24
Delta (response), %	61.1 (11.3)	52.6 (17.0)	0.112	0.60
MDA, nmol/mL				
Baseline	16.2 (9.2)	15.4 (8.9)	0.212	0.04 ^#^
15 min post-TT	18.7 (11.5) *	17.8 (10.1) **	0.427	0.04 ^#^
Delta (response)	2.4 (3.0)	2.4 (2.0)	0.975	0.00 ^#^
Delta (response), %	15.6 (20.4)	14.9 (12.3)	0.942	0.02 ^#^

Notes. Data are mean (SD). Significant effects of the diet are highlighted as bold *p* values and effect sizes. ^#^ Effect size calculated as the partial eta squared (ηp^2^). * *p* = 0.032 compared to baseline; ** *p* = 0.005, *** *p* < 0.001 compared to baseline.

## Data Availability

The raw data presented in this study are available on reasonable request from the corresponding author.

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
