# Peer review of "Effect of Low vs. High Carbohydrate Intake after Glycogen-Depleting Workout on Subsequent 1500 m Run Performance in High-Level Runners"

_nutrients, 2024, doi:10.3390/nu16162763_

Round 1

Reviewer 1 Report

Comments and Suggestions for Authors

The study is well conducted, but the subject matter is not novel. It is well known that CHO restriction reduces performance, especially in events such as the 1500 where the intensity is very close to VO2max, on the other hand, the participants are few and the variety between sexes may alter the results.

INTRODUCTION

The introduction is too short and should explain what is new compared to other CHO supplementation studies, it should be properly justified, this section is very important.

Methods

Participants

Please give more data on the participants in terms of their level, what was their maximal oxygen uptake? 

The sample is small, the authors should explain why no pre-study sample calculation and post-hoc statistical power was performed.

What phase of the menstrual cycle were the women in each of the conditions? This may alter the results.

Please give more information about the training the participants were following at least 3 months before.

Please add a consort Flow diagram and a figure of the study design.

RESULTS

Please represent the results with the main findings in graphs.

Discussion

Well stated, discuss All results and discuss with existing literature.

Author Response

Comments and Suggestions for Authors

The study is well conducted, but the subject matter is not novel. It is well known that CHO restriction reduces performance, especially in events such as the 1500 where the intensity is very close to VO2max, on the other hand, the participants are few and the variety between sexes may alter the results.

Response: We are grateful for the reviewer’s time and thoughtful comments on our manuscript. We have rewritten a significant portion of the Introduction to better highlight the research gaps and the necessity of this study. Additionally, the text has been revised in other sections, a new figure illustrating the study design and main outcomes has been added as graphical abstract (encouraged by the journal), and additional references have been included. Please see below our point-by-point responses to the comments.

INTRODUCTION

The introduction is too short and should explain what is new compared to other CHO supplementation studies, it should be properly justified, this section is very important.

Response: We have rewritten the Introduction as suggested. 

Methods

Participants

Please give more data on the participants in terms of their level, what was their maximal oxygen uptake? 

Response: We described the participants in terms of their training and competitive level. Alas, the testing in the laboratory for VO2max was not conducted for most of the runners of this study, and while the smart watch estimated VO2max was available for some more of the participants, we prefer to not include these data due to questionable validity. However, we added the description of the training for a few months before enrolment into the study.

The sample is small, the authors should explain why no pre-study sample calculation and post-hoc statistical power was performed.

Response: We understand the reviewer‘s concern and have acknowledged this limitation additionally (overall rather small number of participants and low number of female participants) at the end of the discussion in a revised version of the manuscript. For the purpose of the study we preferred to have high recruitment standards in terms of 1500 m run performance to make conclusions applicable to well-trained runners. A sample size of 9-14 participants is common in studies involving high caliber athletes (e.g. Gejl et al. Med Sci Sports Exerc 2014; King et al. Nutrients 2022, PMID: 35565896; Ørtenblad et al.  Eur J Appl Physiol 2024, PMID: 38441690).

What phase of the menstrual cycle were the women in each of the conditions? This may alter the results. 

Response: as indicated in the limitations, the menstrual cycle phase was not checked for, but rather two conditions were separated by 4 weeks, which is roughly the duration of the normal menstrual cycle. While one could agree the phase of the cycle may affect the performance and/or physiological responses, it is much less likely that there is menstrual phase and CHO intake interaction.

Please give more information about the training the participants were following at least 3 months before.

Response: We have added the description of the training for a few months before enrolment into the study.

Please add a consort Flow diagram and a figure of the study design.

Response: We have constructed graphical abstract for to supplement the manuscript with some easy-to-grasp information for a fast overview of the study design.

RESULTS

Please represent the results with the main findings in graphs.

Response: We have included the main findings into the constructed infographic now.

Discussion

Well stated, discuss All results and discuss with existing literature.

Response: There were changes in the Discussion section, and additional references were included. We hope this addresses the changes meant by the reviewer.

Reviewer 2 Report

Comments and Suggestions for Authors

In the introduction section if seems that you highlight evidence that shows a role of supercompensation, but I don't recall reading evidence about depletion on performance.  Is there anything more than just the knee extension study?  
The reason I bring this up is that your hypothesis focuses on restriction; it would be nice to know more about how restriction impacts endurance athletics

There also wasn't any discussion in the intro re: oxidative stress.  Is that warranted to talk more about it?  Maybe not, but would it help to write:

.....oxidative stress.  The purpose of measuring this is to.....

Pg 6

L32: I woudn't say there was a tendency.  It seems the authors are making a stretch here to suggest that there is a trend towards significance.  The sample size is small (did you do an a priori?).  I would change the language here or delete the sentence

Pg 9

Lin 32: change "we"

Create a Conclusion section (instead of having in conclusion in bold)

L16 pg 2 avoid first person language (don't use "we"

L15: add kg/m2 for BMI

Comments on the Quality of English Language

L18: whilst is old style British English; better to use while

Author Response

Comments and Suggestions for Authors

In the introduction section if seems that you highlight evidence that shows a role of supercompensation, but I don't recall reading evidence about depletion on performance.  Is there anything more than just the knee extension study?  
The reason I bring this up is that your hypothesis focuses on restriction; it would be nice to know more about how restriction impacts endurance athletics There also wasn't any discussion in the intro re: oxidative stress.  Is that warranted to talk more about it?  Maybe not, but would it help to write:.....oxidative stress.  The purpose of measuring this is to.....

Response: Thank you for reviewing our manuscript. We have substantially rewritten the Introduction, also based on request by other reviewers. Please see paragraphs 1-5 of the Introduction.

Pg 6

L32: I woudn't say there was a tendency.  It seems the authors are making a stretch here to suggest that there is a trend towards significance.  The sample size is small (did you do an a priori?).  I would change the language here or delete the sentence.

Response: We have changed the language when presenting biomechanical parameters, and we have discussed additionally in the Limitations on the issue of small sample size.

Pg 9

Lin 32: change "we"

Response: We have substituted this with „It has been“.

Create a Conclusion section (instead of having in conclusion in bold)

Response: We have done so now.

L16 pg 2 avoid first person language (don't use "we")

Response: We have checked this throughout and amended where applicable.

L15: add kg/m2 for BMI

Response: Have done so, thank you.

Comments on the Quality of English Language

L18: whilst is old style British English; better to use while

Response: Have replaced, thank you.

Reviewer 3 Report

Comments and Suggestions for Authors

General Comments

I am not sure what is new here. Hi CHO improved performance we have known this for a long time!

Please check work carefully

Abstract

there is a spelling error in the first word, should be carbohydrate

Otherwise abstract is ok in my opinion

Introduction

The Introduction was ok, BUT I thought there was more it could cover. 

This sentence "In addition to racing performance, gross running biomechanics and physiological responses, we also measured an oxidative stress biomarker, plasma malondialdehyde (MDA), an end product of lipid peroxidation, to test whether likely relative glucose deprivation within working muscles under conditions of low CHO availability induced higher oxidative stress." is too long I suggest making it at least two sentences.

Simply a little short so cite a few more references supporting what you did. 

Methods

I have no issues with the Methods. They are well described in an appropriate and scientific fashion. Well done!  

Results

Likewise, I had no issues with the results section. I found it informative and well structured. All of the results made sense and were well presented. Again, good job in my opinion. 

Discussion

The first sentence needs some commas to make it more readable - check throughout please.

You say this "The primary aim of most recent laboratory studies ....." so you need to cite  a few references to support this comment 

I thought there were too many short paragraphs. Many of these could be combined.

Reduce and shorten the limitations section too!  You report what you did - that was fine. We all understand the limitations - I think this is far too long.

On the other hand, I thought you could do more with the conclusions. Maybe include some practical recommendations for athletes here too ! So, simply longer and more detailed. 

Author Response

Comments and Suggestions for Authors

General Comments

I am not sure what is new here. Hi CHO improved performance we have known this for a long time!

Please check work carefully

Response: We are grateful for reviewer’s time and thoughtful comments regarding our manuscript. We have rewritten the large part of the Introduction (in part because of the request from other reviewers) to highlight the gaps in research and the need for the study. Please see paragraphs 1-5 of the Introduction.

Abstract

there is a spelling error in the first word, should be carbohydrate

Response: thanks for spotting, amended now.

Otherwise abstract is ok in my opinion

Introduction

The Introduction was ok, BUT I thought there was more it could cover. 

This sentence "In addition to racing performance, gross running biomechanics and physiological responses, we also measured an oxidative stress biomarker, plasma malondialdehyde (MDA), an end product of lipid peroxidation, to test whether likely relative glucose deprivation within working muscles under conditions of low CHO availability induced higher oxidative stress." is too long I suggest making it at least two sentences.

Simply a little short so cite a few more references supporting what you did. 

Response: We have rewritten the Introduction accordingly. 

Methods

I have no issues with the Methods. They are well described in an appropriate and scientific fashion. Well done!  

Response: Thank you very much for your positive comments.

Results

Likewise, I had no issues with the results section. I found it informative and well structured. All of the results made sense and were well presented. Again, good job in my opinion. 

Response: Thank you very much for your positive comments.

Discussion

The first sentence needs some commas to make it more readable - check throughout please.

Response: We have rephrased the sentese for clarity.

You say this "The primary aim of most recent laboratory studies ....." so you need to cite  a few references to support this comment 

Response: We have cited three of the papers to this end.

I thought there were too many short paragraphs. Many of these could be combined.

Reduce and shorten the limitations section too!  You report what you did - that was fine. We all understand the limitations - I think this is far too long.

Response: We have combined some of the shorter paragraphs of the discussion now and curtailed the limitations slightly. We have condensed the text of the Limitations. However, on request from other reviewers, we elaborated here on the number of the participants recruited to justify the sample size.

On the other hand, I thought you could do more with the conclusions. Maybe include some practical recommendations for athletes here too ! So, simply longer and more detailed.

Response: We have extended the Conclusions section as suggested.

Round 2

Reviewer 1 Report

Comments and Suggestions for Authors

The article is not novel, it is well established that CHO restriction decreases performance. On the other hand, the sample is low and has serious errors such as not controlling for the women's menstrual cycle, which may have altered the results. These facts make it difficult to extrapolate the results, affecting both the internal and external validity of the research. In addition, no performance characteristics of the participants are established with VO2max being estimated. The authors add measurements of MDA but in the introduction they do not justify why this measure should be added. In fact, there is no mention of the effect on oxidative stress.  From my point of view, the study has too many limitations.

On the other hand, the format of the citations does not correspond to that of the journal, these are details that must be taken into account.

Reviewer 2 Report

Comments and Suggestions for Authors

The authors have addressed my prior concerns